# IM-Loss: Information Maximization Loss for Spiking Neural Networks

**Yufei Guo**[*], **Yuanpei Chen**[*], **Liwen Zhang, Xiaode Liu, Yinglei Wang, Xuhui Huang, Zhe Ma**[†]
Intelligent Science & Technology Academy of CASIC
yfguo@pku.edu.cn, rop477@163.com, mazhe_thu@163.com

## Abstract

Spiking Neural Network (SNN), recognized as a type of biologically plausible architecture, has recently drawn much research attention. It transmits information by $0/1$ spikes. This bio-mimetic mechanism of SNN demonstrates extreme energy efficiency since it avoids any multiplications on neuromorphic hardware. However, the forward-passing $0/1$ spike quantization will cause information loss and accuracy degradation. To deal with this problem, the Information maximization loss (IM-Loss) that aims at maximizing the information flow in the SNN is proposed in the paper. The IM-Loss not only enhances the information expressiveness of an SNN directly but also plays a part of the role of normalization without introducing any additional operations (*e.g.*, bias and scaling) in the inference phase. Additionally, we introduce a novel differentiable spike activity estimation, Evolutionary Surrogate Gradients (ESG) in SNNs. By appointing automatic evolvable surrogate gradients for spike activity function, ESG can ensure sufficient model updates at the beginning and accurate gradients at the end of the training, resulting in both easy convergence and high task performance. Experimental results on both popular non-spiking static and neuromorphic datasets show that the SNN models trained by our method outperform the current state-of-the-art algorithms.

## 1 Introduction

Recent progress in Artificial Neural Networks (ANNs) has been successfully applied in many machine intelligence fields. Although ANNs imitate the structure of the brain to a certain extent, their signal processing mechanism using operations on real, dense valued tensors, still differs from how the brain works based on sparse spike events. Being another sub-category of brain-inspired learning models, the Spiking Neural Networks (SNNs) utilize $0/1$ binary spike signals to transmit information between units, and spatio-temporal dynamics to mimic brain behaviors [21; 13; 14]. Unlike the ANN, where all neurons in the same layer need to be evaluated before passing information to the next layer, the SNN processes information asynchronously and sparsely based on sparse spike events. Benefitting from such an information processing paradigm, the SNN can efficiently run on specialized neuromorphic hardware (*e.g.*, Loihi [4]) and has achieved substantial progress in recent years [28].

Although the SNN is more energy-efficient compared with the ANN, it still has some limitations to achieve satisfactory task performance. Due to the usage of spikes for information receiving and transmitting, the SNN needs to quantize real analog values into $0/1$-formed spikes in implementation. This quantization causes information loss in forward propagation. There are some remarkable existing works that alleviate the problem of information loss of SNNs by adjusting the spike rate by learning more parameters. In [26] and [39], a method that aims to learn the correct membrane leak-and-firing threshold for each layer of the SNN was proposed. Results showed that learning the correct threshold

---

[*]Equal Contributions.
[†]Corresponding author.

36th Conference on Neural Information Processing Systems (NeurIPS 2022).

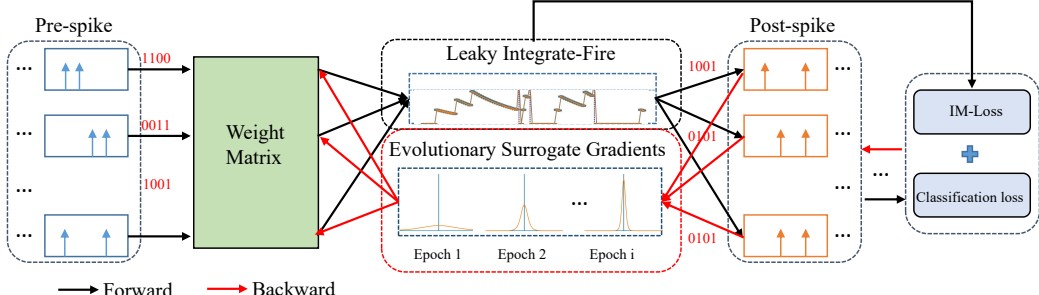

Figure 1: The overall workflow of the proposed algorithm. By minimizing the proposed IM-Loss, the information expressiveness of the SNN can be maximized directly. Meanwhile, the proposed ESG in the backward propagation will automatically evolve in each epoch to behave more like the accurate gradient of the spike activity.

can help increase the information capacity of the SNN. A wise choice of the firing threshold must be small enough that sufficient spikes are fired while also being large enough that avoids all neurons fired at the same time. Similarly, by incorporating the learnable membrane time parameter into the SNN, the parametric neuron [8; 34] can not only increase the diversity of neurons but also the information expressiveness of the network. However, all these methods require more parameters and attempt to increase the information expressiveness of the SNN in an indirect manner. Meanwhile, the non-differentiable spike activities also prevent the SNN from producing accurate gradients in backward propagation, which increase the difficulty of the model optimization.

In this work, we propose information maximization (IM) loss to deal with the information loss in SNNs. More specifically, we first analyze the information expressiveness ability of the SNN from the view of information entropy of the spikes. Then a spike distribution loss is introduced for calculating the information loss, *i.e.*, IM-Loss, of the SNN, thus the information expressiveness of the SNN can be maximized directly by minimizing this information loss, which benefits the accuracy improvement of the SNN. To overcome the non-differentiable spike activities during the model optimization in SNNs, we also introduce Evolutionary Surrogate Gradients (ESG) in the paper. The ESG can evolve automatically during training and provides a pathway to bridge the gap between the pseudo derivative and appropriate derivative. With evolving, it not only ensures sufficient updating at the beginning but also accurate gradients calculation at the end of the training, resulting in networks with better performance. The overall workflow of our algorithm can be seen in Fig.1.

Our contributions can be summarized as follows:

- We propose the IM-Loss to preserve the information in SNNs. It presents a new perspective to understand the ability of the SNN to express information. With the IM-Loss, we can train a more accurate and deeper SNN.

- We also introduce the ESG to effectively approximate the gradients of the spike activity function. The ESG can ensure both sufficient updating of weights at the beginning and accurate gradients at the end of the training in an evolutionary way so that the SNN would be more accurate and easier to converge.

- Our method is evaluated on both static and dynamic datasets. Extensive experimental results show that our algorithm performs remarkably well across various network structures. To our best knowledge, this is the first work that can directly train deep SNNs from scratch without normalization or warm-start techniques.

## 2 Related Work

In this section, we first briefly review the learning methods of SNNs to provide readers an overall understanding. Then we introduce the information theory used in DNNs which gives us insight to design our method for handling information loss in SNNs.

## 2.1 Learning Methods of Spiking Neural Networks

The learning algorithms for SNNs can be divided into three categories: (i) converting ANN to SNN (ANN2SNN) [29; 19]; (ii) unsupervised learning [6]; and (iii) supervised learning [24; 26; 33; 20; 10; 11]. ANN2SNN converts a special trained ANN to an SNN that yields the same input-output mapping for a given task. Since the research of the ANN has been very successful, ANN2SNN can gain competitive accuracy. Some recent conversion methods have achieved nearly loss-less accuracy with VGG and ResNet as original ANN models [19]. However, this kind of method is restricted to rate-coding and can only handle static datasets. Moreover, ANN2SNN requires long timesteps (usually needs hundreds of timesteps) to converge and cannot exploit the spatio-temporal features during training for solving temporal tasks. Unsupervised learning is based on biological plausible local learning rules and is usually considered more biologically plausible than other categories. However, unsupervised learning can only train shallow SNNs and is hard to achieve desirable performance. Supervised learning uses the derivable approximation to overcome the non-differentiability of the spike activities. Hence, the SNN can be optimized with gradient descent algorithms as the ANN and achieves high performance. Moreover, Supervised learning can greatly reduce the number of timesteps and handle dynamic datasets. Thus, supervised learning has increasingly aroused researchers' great interest in recent years. We focus on providing some insights to improve the performance of the supervised learning-based SNNs.

## 2.2 Information Theory in Deep Neural Networks

Information theory is widely used in DNNs, ranging from theoretical investigations (*e.g.*, the generalization bound induced by mutual information [31]) to practical applications (*e.g.*, the variational information bottleneck in representation learning [2]). It provides a useful way to understand the underlying behavior of random variables and this is a key factor in developing and analyzing deep models. There are many popular information-theoretic principles that have demonstrated great potential in DNNs [36]. In [35], deep energy-based models are trained with the $f$-divergence (a generalization of KL divergence) minimization. In reinforcement learning, the maximum entropy regularization is adopted to encourage exploration and avoid getting stuck in a local optima [1]. Mutual information maximization is used in [25] to improve the expressiveness of a model by shifting the distribution of weights to zero-mean in every training iteration. In [3], training DNNs for classification is investigated via minimizing the information bottleneck functional. Despite the great success of information theory in DNNs, we find no related works in supervised learning of SNNs. In the paper, we propose the IM-Loss, which is based on the information maximization principle, to improve SNNs. Our implementation and idea of the information maximization are very different from these existing works, which may also provide a new idea to further improve these methods.

# 3 Preliminary

For better expression, we will first introduce the iterative Leaky Integrate-and-Fire (LIF) model and surrogate gradients of SNNs in detail, which are also adopted in our method.

## 3.1 Explicitly Iterative LIF Model

The LIF neuron is a commonly used mathematical model to describe the behavior of neural activities, which includes spike firing and update of membrane potential. Formally, a LIF neuron in layer $l$ of an SNN with index $i$ can be given by

$$\tau \frac{\partial u_i^l}{\partial t} = -u_i^l + I_i^l, \quad u_i^l < V_{th} \tag{1}$$

$$fire\ a\ spike \quad \& \quad u_i^l = u_{rest}, \quad u_i^l \geq V_{th} \tag{2}$$

where $u_i^l$ is the membrane potential, $u_{rest}$ is the membrane resting potential, $\tau$ is the membrane time constant to describe the membrane potential decaying, $I_i^l$ is the pre-synaptic input, and $V_{th}$ is a given firing threshold. From Eq. 1, we can see that $u_i^l$ actually acts as a leaky integrator of the pre-synaptic input $I_i^l$. And Eq. 2 shows that when a neuron membrane potential reaches the firing threshold $V_{th}$, it emits a spike to transfer its information to other neurons and reset its membrane potential to $u_{rest}$.

The above differential expressions described in the continuous domain are difficult to be implemented on mainstream machine learning frameworks (*e.g.*, Pytorch, Tensorflow), where the embedded automatic differentiation mechanism executes discretely. Alternatively, the iterative LIF model [33] used the Euler method to solve the first-order differential equation of the LIF model. The iterative expression of the LIF model can be governed by

$$u_i^l(t) = \tau_{decay}u_i^l(t-1) + I_i^l(t), \quad u_i^l(t) < V_{th} \tag{3}$$

$$fire\ a\ spike \quad \& \quad u_i^l(t) = u_{rest}, \quad u_i^l(t) \geq V_{th} \tag{4}$$

where $\tau_{decay}$ is a constant to describe membrane potential decaying, $u_i^l(t)$ is the membrane potential in timestep $t$, and $I_i^l(t)$ is the pre-synaptic input. Since $I_i^l(t)$ is the accumulation of spikes from the neurons of layer $l-1$ connected to the current neuron, it can be described as $I_i^l(t) = \sum_j w_{ij}o_j(t)$. Where $w_{ij}$ is the weight of $j$-th pre-synaptic of the current neuron, $o_j(t)$ is the output of the neuron from the previous layer, which interacts on the current neuron's $j$-th pre-synaptic at the moment of $t$. If we set $u_{reset} = 0$, the whole iterative LIF model in both spatial and temporal domain can be determined by

$$u_i^l(t) = \tau_{decay}u_i^l(t-1)(1 - o_i^l(t-1)) + \sum_j w_{ij}o_j(t) \tag{5}$$

$$o_i^l(t) = \begin{cases} 1, & if\ u_i^l(t) \geq V_{th}, \\ 0, & otherwise. \end{cases} \tag{6}$$

The iterative LIF model is friendly to be implemented on general machine learning programming frameworks.

## 3.2 Surrogate Gradients of SNNs

The spike activity function shown as Eq.6 can be viewed as a variant of the sign function. Its derivative is 0 everywhere except for the infinity value at $V_{th}$. Since the non-differentiability of spike activity function, the back-propagation cannot be applied directly for training SNN. To demonstrate this challenge, we show the expression to compute gradients of weights obtained by the spatial-temporal back-propagation (STBP) as follows,

$$\Delta\mathbf{W} = \frac{\partial\mathcal{L}}{\partial\mathbf{W}} = \sum_{t=1}^T \frac{\partial\mathcal{L}}{\partial\boldsymbol{y}^t}\frac{\partial\boldsymbol{y}^t}{\partial\boldsymbol{u}^t}\frac{\partial\boldsymbol{u}^t}{\partial\mathbf{W}} \tag{7}$$

where $\boldsymbol{y}$ denotes the target output vector, $\mathbf{W}$ is the matrix of weights of the SNN, $\mathcal{L}$ is the loss function, and $\frac{\partial\boldsymbol{y}^t}{\partial\boldsymbol{u}^t}$ is the gradient of the spike activity function. The weight update is computed as

$$\mathbf{W} = \mathbf{W} - \eta\Delta\mathbf{W} \tag{8}$$

where $\eta$ is the learning rate. Considering that, $\frac{\partial\boldsymbol{y}^t}{\partial\boldsymbol{u}^t}$ is either 0 almost everywhere or infinity at $V_{th}$, $\Delta\mathbf{W}$ is either 0 everywhere or a very large value in a rare case, hence weights would likely not be changed or updated to a large value some times.

Most prior works adopted surrogate gradients (SG) to overcome the non-differentiability challenge [38; 17; 32; 23; 30]. However, the SGs in existing methods are fixed during training, such a strategy would limit the learning capacity of the network. We notice that the SG methods are also adopted in Quantization Neural Networks (QNNs) which also suffer from non-differentiability. Some changeable surrogate gradient methods that can achieve better performance by changing SG in the training are proposed in QNNs [25; 9]. Combining characteristics of SNNs, an evolutionary SG that can maintain strong updating ability in the early training phase and approach the accurate gradients gradually at the end of the training is introduced in the paper.

## 4 Methodology

In this paper, we argue that the challenge of training highly accurate spiking neural networks is mainly limited by the severe information loss caused by the spike activity function. To mitigate the problem, the IM-Loss is proposed. Furthermore, we also introduce the ESG to obtain the appropriate surrogate gradient for the non-differentiable spike activity in backward propagation.

## 4.1 Information Maximization Loss

In the forward propagation, the membrane potential to $0/1$ spikes quantization of the spike activity function will cause information loss unquestionably. This is one of the main reasons why an SNN model cannot achieve as satisfactory performance as its ANN counterpart. Ideally, the spike tensor $\mathbf{O}$ should reflect the information of the membrane potential tensor $\mathbf{U}$ as much as possible. In the view of the information flow, to maximize the information flow from the full-precision tensor $\mathbf{U}$ to the binary tensor $\mathbf{O}$, the mutual information $I(U; O)$ of the random variables $U$ and $O$ should be maximized as follows,

$$\arg\max_{U,O} I(U; O) = H(O) - H(O|U) \tag{9}$$

where $H(O)$ is the entropy of $O$ and $H(O|U)$ is the conditional entropy of $O$ given $U$. It can be proved that, maximizing the mutual information $I(U; O)$ is equivalent to maximize the information entropy $H(O)$ for SNNs (see Appendix A.1 for detailed proofs) as follows,

$$\arg\max_{U,O} I(U; O) = H(O). \tag{10}$$

Obviously, when the distribution of $O$ satisfies $p(0) = p(1) = 0.5$ the entropy $H(O)$ is maximized, where the $p$ is the marginal probability mass functions of the discrete variables $O$. This means that to maximize the information flow from the full-precision tensor $\mathbf{U}$ to the spike tensor $\mathbf{O}$, the spike values should be evenly distributed, $i.e.$, $\sum_{u<V_{th}} p(u) = \sum_{u \geq V_{th}} p(u) = 0.5$. We set $U_{(q)}$ to denotes the $q$-quantile of $U$'s elements in ascend order where $0 \leq q \leq 1$. To maximize the $H(O)$, we should have $U_{(0.5^-)} < V_{th}$ and $U_{(0.5^+)} \geq V_{th}$. Since the $U$ follows a Gaussian distribution [38; 10], $U_{(0.5)}$ is almost equal to the mean value $\bar{U}$. Then $(\bar{U} - V_{th})^2$ can be regarded as a metric of the information-loss caused by the quantization from the full-precision tensor $\mathbf{U}$ to the spike tensor $\mathbf{O}$. Hereby, we define the proposed IM-Loss as:

$$\mathcal{L}_{IM} = \sum_{l=0}^{L} (\bar{U}_l - V_{th})^2 / L \tag{11}$$

where $\bar{U}_l$ is the average membrane potential at all timesteps of $l$-th layer, and $L$ is the total number of layers. Then the overall loss can be calculated as follows,

$$\mathcal{L}_{Total} = \mathcal{L}_{CE} + \lambda \mathcal{L}_{IM} \tag{12}$$

where $\mathcal{L}_{CE}$ is the cross-entropy loss, and $\lambda$ is a coefficient to balance the classification loss and the information loss. We set $\lambda$ as 2 in this paper.

## 4.2 The Effect of IM-Loss

SNNs utilize $0/1$ spikes to communicate between neurons. Benefitting from this information processing paradigm, SNNs run efficiently. However, since the spike rate of the network with random weights decreases sharply through layers, an SNN is difficult to train. In other words, the spike rate will usually soon become 0 after several layers when training models from scratch, resulting in information flow disappearing and difficulty for training large and deep SNNs.

For better understanding, the layer-wise average spike rates of spiking CIFARNet and ResNet-19 initialized with random weights on the CIFAR10 test set are visualized in Fig. 2. Both networks were trained with a timestep of 4 and without normalization. When epoch is 0, $i.e.$, the untrained networks, spike rates are becoming 0 in the deeper layers, resulting in untrainable networks. This means deep SNNs are not easy to train without any special techniques. Some works attempted to alleviate this phenomenon by using normalization techniques. However, normalization requires other additional operations. For instance, the bias operation consumes extra computations in the inference phase when implementing models on some specialized neuromorphic hardware. While for the case of epoch=1, where the proposed IM-Loss was applied, the spike rate in each layer can be restricted to an appropriate range ($i.e.$, about $50\%$), which enables the learning algorithm to train a large and deep SNN. By maintaining an appropriate range for the spike rate, IM-Loss can effectively update the weights of the two deep SNNs even with random initialization. In this sense, IM-Loss also possesses the effect of normalization but without other additional operations or well-designed initialization strategies.

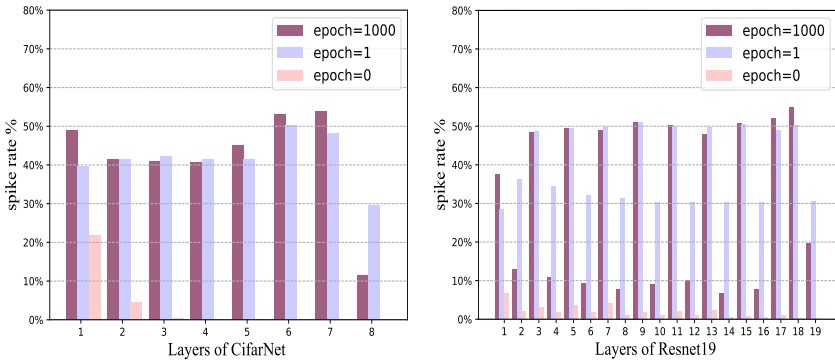

Figure 2: Layer-wise spike rate for CIFARNet (left) and ResNet-19 (right) during inference over entire CIFAR10 test-set with epoch = 0, 1 and 1000.

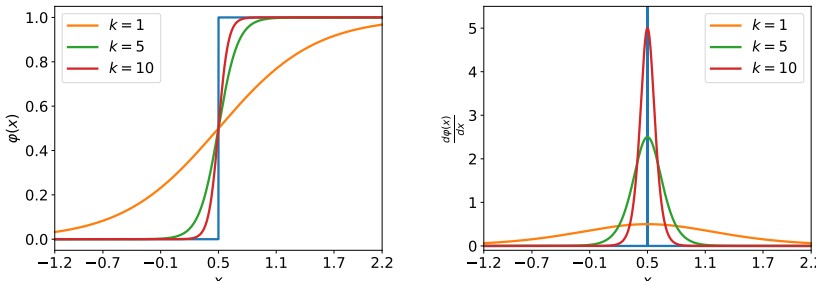

Figure 3: The response curves of asymptotic function (left) and its corresponding gradient (right), under different values of the coefficient, $k$. The blue curves represent the spike activity function (left) and the true gradient of the spike activity (right).

To further demonstrate the information retention ability of IM-Loss, the case of epoch=1000 in Fig. 2 shows the spike rate of the SNNs at the end of training. For ResNet-19, IM-Loss did not restrict the layers inside the residual blocks, since these layers tend to learn small perturbations which are incompatible with the restriction by IM-Loss. It can be seen that the spike rate of each layer restricted by IM-Loss is near 0.5, meaning the information flow can be maximized by IM-Loss as analyzed above.

### 4.3 Evolutionary Surrogate Gradients

The zero-derivative of spike activity function almost everywhere extremely prevents SNNs from training adequately. To our best knowledge, nearly all the prior works attempted to alleviate this problem by following the principle of replacing the gradient of the spike activity function with a fixed SG in SNNs. As aforementioned, changeable surrogate gradient (CSG) methods in QNNs can achieve better performance. We guess that designing a suitable CSG can also increase accuracy in SNNs. To this end, we first introduce a differentiable asymptotic function to approximate the spike activity function as follows,

$$\varphi(x) = \frac{1}{2}tanh(k(x - V_{th})) + \frac{1}{2} \tag{13}$$

where the coefficient $k$ determines the shape of the asymptotic function. The gradient of the asymptotic function, $\varphi(\cdot)$ is used to replace the gradient of the spike activity function.

For the sake of intuitive explanation, Fig. 3 illustrates the influences of $k$ on the asymptotic function (left) and its corresponding gradient (right). It can be seen that, when setting $k$ with a large value, a more accurate gradient in backward can be obtained, so as to increase the difficulty of the training. Conversely, when $k$ becomes smaller, the gradient of the asymptotic function will tend to be uniform and keep some relatively large value, thus the strong weight updating ability can be obtained.

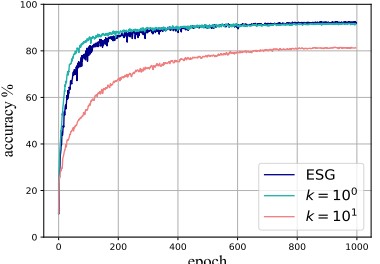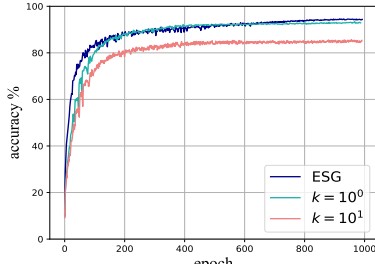

Figure 4: The accuracy curves of CIFARNet (left) and ResNet-19 (right) with $k = 10^0$, $k = 10^1$ and ESG. The ESG method maintains the updating ability at the early stage and progressively reduces the error to achieve high accuracy.

Then we design the **ev**olutionary **a**symptotic **f**unction (EvAF) as follows,

$$\varphi(x) = \frac{1}{2}tanh(K(i)(x - V_{th})) + \frac{1}{2} \tag{14}$$

where $\varphi(x)$ is the backward approximation substitute for the spike activity function and $K(i)$ is the coefficient that changes dynamically with the training epoch as follows,

$$K(i) = \frac{(10^{\frac{i}{N}} - 10^0)K_{max} + (10^1 - 10^{\frac{i}{N}})K_{min}}{9} \tag{15}$$

where $K_{min}$ and $K_{max}$ denotes the lower bound and the upper bound of the value range of $K$, respectively, and $i$ is the index of epoch starting from 0 to $N - 1$. In practice, we set $K_{min} = 10^0$ and $K_{max} = 10^1$. Without using any additional hyper-parameter, $K(i)$ will increase with the accumulation of training epochs. Driven by $K(i)$, EvAF can gradually evolve to the spike activity function, and a more accurate gradient will be obtained, correspondingly. Such dynamically changing gradient will be used as the SG in our method, namely Evolutionary Surrogate Gradient (ESG). By introducing the modified asymptotic function, EvAF into the SNNs, both weight updating ability in the early stage of training and more accurate backward gradient at the end of training can be fulfilled by the proposed ESG.

### 4.4 The Effect of ESG

To demonstrate the effect of ESG, we compared the accuracy curves of spiking CIFARNet and ResNet-19 (with timestep = 4) on CIFAR10 using the gradient of the asymptotic function with $k = 10^0$, $k = 10^1$ and ESG derived from EvAF with $K \in [10^0, 10^1]$ in backward. From the visualized comparison results shown in Fig. 4, we can see that the SNNs with a small-valued $k$ have stronger updating ability and are easier to converge compared to the SNNs with a large $k$. The SGs of SNNs with a large $k$ are very close to the gradients of the spike activities, in such case, most weights of these SNNs cannot be updated sufficiently. Hence these SNNs will suffer slow convergence and face severe performance degradation. Our ESG starts with $K_{min}$ at the early stage of training, so that the strong updating ability can make the SNN converge easily; then it ends with $K_{max}$ at the end of training thus the backward gradient will be more accurate resulting in a better-performed SNN. In general, by using a dynamically changing coefficient, our ESG encourages the SNN with both easier convergence and higher accuracy.

## 5 Experiments

In this section, extensive experiments were conducted to demonstrate the effectiveness of the proposed methods. The algorithm of the training process of our method is presented in Appendix A.2. The experiments include widely-used network structures including spiking CIFARNet [33], ResNet-19 [38], modified VGG-16 [26], and ResNet-34 [12] for both popular non-spiking static and neuromorphic datasets: CIFAR10/100, ImageNet (ILSVRC12) and CIFAR10-DVS [18]. For being friendly with neuromorphic hardware, the max-pooling layer was replaced with the average-pooling layer in the used network architectures.

Table 1: Ablation experiments for IM-Loss and ESG.

| Architecture | Methods | Accuracy |
|---|---|---|
| CIFARNet | w/o normalization | 89.83% |
| | w/ tdBN [38] | 90.69% |
| | w/ IM-Loss | 90.90% |
| | w/ ESG | 91.17% |
| | w/ IM-Loss & ESG | 91.75% |
| | w/ IM-Loss, ESG & tdBN | **92.20%** |
| ResNet-19 | w/o normalization | 91.23% |
| | w/ tdBN [38] | 92.92% |
| | w/ IM-Loss | 94.29% |
| | w/ ESG | 94.44% |
| | w/ IM-Loss & ESG | 94.64% |
| | w/ IM-Loss, ESG & tdBN | **95.40%** |

## 5.1 Ablation Study

To understand how our method works in practice, the ablation experiments for IM-Loss and ESG with spiking CIFARNet and ResNet-19 (timestep = 4) were first conducted on CIFAR10. To make a fair comparison with tdBN, the rectangular surrogate function was used for the model without ESG to compute gradients, as in tdBN. The performances of the networks with IM-Loss and ESG alone and their combinations are shown in Table 1. For the networks without normalization or IM-Loss, there is a certain probability that they might not be trainable from scratch with inappropriate random initialization as aforementioned. Therefore, in this series of experiments, the models trained by IM-Loss after several iterations (training was stopped when the spike rates of the last layer were not 0) as the initialization, which can also be regarded as a warm-start for the networks. Results in table 1 show that both IM-Loss and ESG can improve the networks' performances alone. And by using the combination of these two methods, the performances can be further improved. Moreover, our method is also compatible with other advanced technologies, such as tdBN [38], hence it can be applied in a plug-in manner to enhance the learning capacity of SNNs together with other techniques.

## 5.2 Comparisons with State-of-the-art Methods

As reported in the ablation study, our algorithm combined with tdBN can achieve better results, hence this training technique was also adopted in our method. The results are listed in table 2.

On CIFAR10, our models achieve better performances than the other state-of-the-art methods (92.20% top-1 accuracy with only 4 timesteps by CIFARNet, 93.85% top-1 accuracy with only 5 timesteps by VGG-16, and 95.49% top-1 accuracy with only 6 timesteps by ResNet-19). Moreover, with only 2 timesteps, our method using ResNet-19 as backbone still outperforms the STBP-tdBN with 6 timesteps by 0.69% accuracy. And our CIFARNet-based model with only 4 timesteps can also outperform the CIFARNet in [33] with 12 timesteps and the one in [37] with 5 timesteps by 1.67% and 0.79% accuracy, respectively. These comparison results clearly show the effective latency reducing the ability of our method. And the same situation can also be seen from the results on a more complex dataset, CIFAR100.

ResNet-34 and VGG-16 were used as backbones on ImageNet. Our ResNet-34 and VGG-16 achieve 67.43% and 70.03% top-1 accuracy with only 6 and 5 timesteps, respectively. Compared to STBP-tdBN and TET, our ResNet-34 obtains 3.71% and 2.64% accuracy improvement, respectively. Compared to Diet-SNN [26] , our VGG-16 obtains 1.65% accuracy improvement. The Dspike is some better than our method. However, the Dspike is very time-consuming than our method since it is computed by finite difference manner. Evaluating finite difference could be time-consuming, where for each single weight, the model should run two times to evaluate the difference of the loss, and a model can have more than ten million parameters.

To further verify the effectiveness of our methods, the neuromorphic dataset, CIFAR10-DVS, was also used to evaluate the proposed method. With ResNet-19 as the backbone, it achieves the best performance with 72.60% accuracy in 10 timesteps, which is 4.80% higher than STBP-tdBN.

Table 2: Performance comparisons of our method and other state-of-the-art networks.

| Dataset | Method | Type | Architecture | Timestep | Accuracy |
|---|---|---|---|---|---|
| CIFAR10 | SpikeNorm [29] | ANN2SNN | VGG-16 | 2500 | 91.55% |
| | Hybrid-Train [27] | Hybrid | VGG-16 | 200 | 92.02% |
| | Spike-basedBP [16] | SNN training | ResNet-11 | 100 | 90.95% |
| | STBP [33] | SNN training | CIFARNet | 12 | 90.53% |
| | TSSL-BP [37] | SNN training | CIFARNet | 5 | 91.41% |
| | Diet-SNN [26] | SNN training | VGG-16 | 5 | 92.70% |
| | PLIF [8] | SNN training | PLIFNet | 8 | 93.50% |
| | Dspike [20] | SNN training | ResNet-18 | 6 | 94.25% |
| | STBP-tdBN [38] | SNN training | ResNet-19 | 6 | 93.16% |
| | | | | 4 | 92.92% |
| | | | | 2 | 92.34% |
| | TET [5] | SNN training | ResNet-19 | 6 | 94.50% |
| | | | | 4 | 94.44% |
| | | | | 2 | 94.16% |
| | **Our method** | SNN training | ResNet-19 | 6 | **95.49%**±0.05 |
| | | | | 4 | **95.40%**±0.08 |
| | | | | 2 | 93.85%±0.10 |
| | | | VGG-16 | 5 | **93.85%**±0.11 |
| | | | CIFARNet | 4 | **92.20%**±0.12 |
| CIFAR100 | BinarySNN [22] | ANN2SNN | VGG-15 | 62 | 63.20% |
| | Diet-SNN [26] | SNN training | VGG-16 | 5 | 69.67% |
| | **Our method** | SNN training | VGG-16 | 5 | **70.18%**±0.09 |
| ImageNet | SpikeNorm [29] | ANN2SNN | ResNet-34 | 2500 | 65.47% |
| | SEW ResNet [7] | SNN training | ResNet-34 | 4 | 67.04% |
| | STBP-tdBN [38] | SNN training | ResNet-34 | 6 | 63.72% |
| | Diet-SNN [26] | SNN training | VGG-16 | 5 | 69.00% |
| | TET [5] | SNN training | ResNet-34 | 5 | 64.79% |
| | Dspike [20] | SNN training | ResNet-34 | 6 | **68.19**% |
| | | | VGG-16 | 5 | **71.24**% |
| | **Our method** | SNN training | ResNet-34 | 6 | 67.43%±0.11 |
| | | | VGG-16 | 5 | 70.65%±0.07 |
| CIFAR10-DVS | Rollout [15] | Streaming | DenseNet | 10 | 66.80% |
| | STBP-tdBN [38] | SNN training | ResNet-19 | 10 | 67.80% |
| | **Our method** | SNN training | ResNet-19 | 10 | **72.60%**±0.08 |

# 6   Conclusion

This work aims at addressing two main problems for training accurate and deep SNNs: (i) the information flow loss caused by the 0/1 spike quantization; (ii) the training obstacle caused by the non-differentiable spike activity function. Inspired by the correlation between membrane potentials and binary spike signals from the view of information entropy, we design a novel loss, IM-Loss, which can directly maximize the information expressiveness of an SNN, so that a well-performed SNN can be obtained. It also plays part of the normalization's role but does not require additional operations or well-designed initialization strategies. Then, by incorporating a dynamically changing coefficient into a differentiable asymptotic function, we introduce the ESG method for training SNN in a more appropriate way. ESG encourages the SNN to enjoy both easy convergence and high accuracy by adjusting the weight updating ability and the accuracy of the gradients. Combining these two methods, our method outperforms the state-of-the-art methods.

# Acknowledgment

This work is supported by grants from the National Natural Science Foundation of China under contracts No.12202412 and No.12202413.

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
