# IM-Loss: Information Maximization Loss for Spiking Neural Networks

**Yufei Guo**[*], **Yuanpei Chen**[*], **Liwen Zhang, Xiaode Liu, Yinglei Wang, Xuhui Huang, Zhe Ma**[†]
Intelligent Science & Technology Academy of CASIC
yfguo@pku.edu.cn, rop477@163.com, mazhe_thu@163.com

## A  Appendix

### A.1  Proofs of Zero Conditional Entropy

*Proof.*

$$I(U;O) = H(O) - H(O|U) \tag{1}$$

$$= \sum_{u,o} p(u,o)log\frac{p(u,o)}{p(u)p(o)} \tag{2}$$

$$= \sum_{u,o} p(u,o)log\frac{p(u,o)}{p(u)} - \sum_{u,o} p(u,o)logp(o) \tag{3}$$

$$= \sum_{u,o} p(u)p_{O|U=u}(o)logp_{O|U=u}(o) - \sum_{u,o} p(u,o)logp(o) \tag{4}$$

$$= \sum_{u} p(u)(\sum_{o} p_{O|U=u}(o)logp_{O|U=u}(o)) - \sum_{o}(\sum_{u} p(u,o))logp(o) \tag{5}$$

$$= -\sum_{u} p(u)H(O|U=u) - \sum_{o} p(o)logp(o) \tag{6}$$

$$= -H(O|U) + H(O) \tag{7}$$

In above equations, $p(u)$, $p(o)$ and $p(u,o)$ are the marginal probability mass functions of the discrete variables $U$, $O$ and their joint probability mass function. The conditional entropy $H(O|U)$ can be expressed as the below equation according to the Eq.5 and Eq.7.

$$H(O|U) = \sum_{u} p(u)(\sum_{o} p_{O|U=u}(o)logp_{O|U=u}(o)) \tag{8}$$

Since every $u$ is corresponding to a fixed spike 0 or 1, $p_{O|U=u} = 0$ or 1. So we have

$$H(O|U) = \sum_{u} p(u)(0 + 0 + \cdots + 0) = 0 \tag{9}$$

Then maximizing the mutual information $I(U;O)$ is equivalent to maximizing the information entropy $H(O)$:

---

[*]Equal Contributions.
[†]Corresponding author.

36th Conference on Neural Information Processing Systems (NeurIPS 2022).

$$\arg\max_{U,O} I(U;O) = H(O) \tag{10}$$

## A.2 Algorithm

The proposed training algorithm of an SNN is presented in Algo.1.

---

**Algorithm 1** The proposed training algorithm of an SNN.

---

**Input**: Initialized SNN; training dataset; total training epochs, $I$; training iterations per epoch, $J$.
**Output**: The trained SNN.

1: **for** all $i = 1, 2, \ldots, I$-th epoch **do**
2:     **for** all $j = 1, 2, \ldots, J$-th iteration **do**
3:         **Forward propagation:**
4:         Compute classification loss $\mathcal{L}_{CE}^j$.
5:         **for** all $l = 1, 2, \ldots, (L-1)$-th layers (IM-Loss is not added for the last output layer.) **do**
6:           Compute $\bar{\mathbf{U}}_l$ for $l$-th layer.
7:         **end for**
8:         Compute IM-Loss $\mathcal{L}_{IM}^j$.
9:         Compute overall loss $\mathcal{L}_{Total}^j$.
10:        **Back propagation:**
11:        Update the $g'(\cdot)$ via ESG:
12:        $g'(x) = \frac{1}{2}K(i)(1 - tanh(K(i)(x - V_{th}))^2)$
13:        Calculate the gradients w.r.t. $\mathbf{W}$:
14:        $\frac{\partial \mathcal{L}_{Total}^j}{\partial \mathbf{W}} = \sum_{t=1}^{T} \frac{\partial \mathcal{L}_{Total}^j}{\partial y^t} g'(u^t) \frac{\partial u^t}{\partial \mathbf{W}}$, where $y^t$ and $u^t$ denote the target output and membrane potential at $t$-th timestep.
15:        **Parameters Update**
16:        Update $\mathbf{W} : \mathbf{W} = \mathbf{W} - \eta \frac{\partial \mathcal{L}_{Total}^j}{\partial \mathbf{W}}$, where $\eta$ is learning rate.
17:     **end for**
18: **end for**
19: **return** the trained SNN.

---