# OpenReview forum: "IM-Loss: Information Maximization Loss for Spiking Neural Networks"
_NeurIPS.cc/2022/Conference — NeurIPS 2022 Accept_

### Official Review · Reviewer_jVTP · 2022-07-08

**Rating:** 6
**Confidence:** 5
**Soundness:** 3 good
**Presentation:** 4 excellent
**Contribution:** 3 good

**Summary:**

The paper focuses on improving spiking neural networks' performance, and presents two major contributions:

### 1. IM-loss

The information maximization loss (IM-loss) is evaluated by the $l_2$ distance between membrane potential and the neuron threshold. This form is based on three assumptions:

- Maximizing the information flow between membrane potential and output spike train can benefit SNNs' accuracy performance.

- The information flow is maximized when the spike trains' probability of 0 and 1 are equal.

- The membrane potential follows the Gaussian distribution, which implies the distribution's mean value is roughly equal to the distribution's half probability point (cumulative distribution function = 0.5).

### 2. ESG

The evolutionary surrogate gradient (ESG) adjusts the shape of the surrogate gradient function along the training process. The strategy is based on two assumptions:

- The wider shape of the surrogated gradient of the activation function, the stronger weight updating ability.

- The more accurate (sharp) of the gradient, the better training accuracy can be reached.



**Questions:**

1. Can you prove that the proposed regularizer can achieve a better trade-off between spiking sparsity and accuracy?

2. If one includes the temporal dependency into consideration, will the results of information flow maximization change?

3. What does the membrane potential distribution looks like in your experiment? Does it approximately follow the Gaussian distribution with a free-to-change mean value?

4. How do you evaluate the difference between ESG and  [2] mentioned above?


**Limitations:**

Yes.

**Strengths And Weaknesses:**

### Strengths

The paper is well-drafted and presents the ideas clearly. The ideas are well evaluated experimentally.

### Weaknesses

1. Novelty.

- Adding homeostatic mechanisms to regularize spiking neuron's firing activity is not a new trick. Previously, [1] adds a regularizer to drive SNNs' firing activity more sparse to save energy. However in this work, the author proposed regularizer drives membrane potential near threshold, which potentially increases the firing activity and leads to poorer energy efficiency.

- Updating the shape of the surrogated function is also not new. [2] proposed to automatically adjust the temperature of the surrogated activation function in order to achieve the best training performance. Comparing to this work, [2] is more flexible.

2. Soundness

- The theoretical reasons (see summary above) the authors proposed that lead to the IM-loss are questionable. First, information loss is necessary for classification tasks (The whole image is compressed through a neural network and leaves its classes information only in the output layer). Therefore we do not know whether maximizing informaton flow benefits the performance. Second, the measurement of information flow neglects the important temporal dependency of LIF neurons, so the result that p(0)=p(1)=0.5 maximizes the information flow based on elementary information theory is oversimplified. Third, the membrane potential actually does not follow the Gaussian dist with free to change mean value. The paper cites [3] here to back up their claim, yet [3] only mentioned that the membrane potential distribution approximately follows a *zero mean* Gaussian distribution.

- The heuristic description of the ESG is also questionable. How do the authors evaluate the "weight updating ability"? I understand that too sharp a surrogated gradient function cannot sustain training, but the function is also not "the wider the stronger". Too wide the function is also harmful for training. Besides, it is also dubious to claim adjusting the function more and more accurate benefits the training of SNNs. Please refer to discussions in [2] [4] [5].


[1] Zenke F, Ganguli S. Superspike: Supervised learning in multilayer spiking neural networks[J]. Neural computation, 2018, 30(6): 1514-1541.

[2] Li Y, Guo Y, Zhang S, et al. Differentiable spike: Rethinking gradient-descent for training spiking neural networks[J]. Advances in Neural Information Processing Systems, 2021, 34: 23426-23439.

[3] Zheng H, Wu Y, Deng L, et al. Going deeper with directly-trained larger spiking neural networks[C]//Proceedings of the AAAI Conference on Artificial Intelligence. 2021, 35(12): 11062-11070.

[4] Zenke F, Vogels T P. The remarkable robustness of surrogate gradient learning for instilling complex function in spiking neural networks[J]. Neural computation, 2021, 33(4): 899-925.

[5] Yang Y, Zhang W, Li P. Backpropagated neighborhood aggregation for accurate training of spiking neural networks[C]//International Conference on Machine Learning. PMLR, 2021: 11852-11862.

---

> ### Author Response · Authors · 2022-08-02
> **Response to Reviewer jVTP (Part V)**
>
> Other questions.
>
> ---
>
> Q5: Can you prove that the proposed regularizer can achieve a better trade-off between spiking sparsity and accuracy?
>
> A5: Thanks for your constructive advice. Here, we provide some efficiency analysis. We define the average spike rate as total #$spikes/#neurons on the test-set. For tdBN, the avg-rate of ResNet-19 on CIFAR10 are 0.35, 0.67, 0.82 with T = 2, 4, 6. while 0.81, 1.59, 2.37 for IM-Loss. However, with only T=2, our method outperforms the tdBN with T=6 by 0.69\% accuracy with a low avg-rate. IM-Loss increases the fire rate but reduces the timesteps at the same time, i.e., the proposed regularizer can achieve a better trade-off between spiking sparsity and accuracy. On the other hand, SNNs can also run on GPUs. In this situation, the fewer timesteps, the better, and our method enjoys very few timesteps.
>
> ---
>
> Q6: If one includes the temporal dependency into consideration, will the results of information flow maximization change?
>
> A6: Thanks for your constructive advice. we advocate for your view that the information flow is temporal dependent. However, this will still not change the fact the information flow will be maximized when  p(0)=p(1)=0.5 based on the information theory we adopted. Though the p(0)=p(1)=0.5 is seen as simple, it is the result derived from the theory but not the implementation manner of the theory. Considering the temporal dependency  of LIF neuron is very complex and difficult to express explicitly and uniformly, we choose optimizing spiking rate only based on the information theory and do not design any explicit temporal dependency module in the IM-Loss. In this way, their dependency can be learned freely and implicitly in the training instead of manual design beforehand. This is like we do image classification task with learned CNNs instead of handcrafted descriptors. Furthermore, this design is very simple and general thus easy to be applied to other models.
>
> ---
>
> Q7: What does the membrane potential distribution looks like in your experiment? Does it approximately follow the Gaussian distribution with a free-to-change mean value?
>
> A7: I'm sorry for this neglect of experiment proof. In our experiment, the membrane potential indeed follow the Gaussian-alike distribution with free mean value. We provide this experiment proof in the appendixes section of revised manuscript for you checking. tdBN [3] also follow the Gaussian-alike distribution with free mean value in our experiments and other work (see Fig. 2 in [8]).
>
> [8] Guo, Y., Tong, X., , et al. Recdis-snn: Rectifying membrane potential distribution for directly training spiking neural networks. In: Proceedings of the IEEE/CVF Conference on Computer Vision and Pattern Recognition (CVPR2022). pp. 326–335 (June 2022)
>
> ---
>
> Q8: How do you evaluate the difference between ESG and [2] mentioned above?
>
> Different methods are adoptted in our work and [2]. Here, we want to provide some discrepancies between the two methods. Our method is designed manually, and in our experiments, the SNN with ESG is better than the vanilla SNN but will induce no extra computation in the training. While the Dspike[2] is computed by finite difference method, however, evaluating finite difference could be time-consuming, since for each single weight, the model should run two times to evaluate the difference of the loss, and a model can have more than ten million parameters (e.g. 11,200,000 parameters for ResNet18), which greatly slows down the training process. To reduce the computation, the Dspike[2] chooses to only compute the finite difference in the first layer to represent the surrogate gradients of the whole model. However, this is still very time-consuming. Take ResNet20 in Cifar10 for example as introduced in [2], if we set batch-size as 128. The computation for the finite difference in the first layer once is equal to ResNet20 inferring about 4.5 epochs on training set. To sum up, the Dspike performs remarkably while is very time-consuming. Our method is more efficient with a relatively good performance. Hence, we think ESG function and Dspike function are both meaningful for SNN field. This is some like that SGD optimizer is designed manually from experience, while Meta-optimizer is learned by learning[6] from optimal scheme. Through Meta-optimizer[6] performs better than SGD optimizer, it is time-consuming and more complex to use. Hence it cannot be concluded that Meta-optimizer is better and more flexible than SGD optimizer.

---

> > ### Comment · Reviewer_jVTP · 2022-08-05
> > **Respond to Part V**
> >
> > Q5.  CIFAR10 is a static image dataset, so no temporal dependency is required. Even T=1 is sufficient to achieve successful training and achieve high accuracy. In my view, the results, on the contrary, prove that the proposed regularizer increased accuracy simply by increasing the spiking rate.
> >
> > Q6. Q7. See my response above.
> >
> > Q8. Thanks for your reply. The heuristically designed ESG function has a clean intuitive understanding. The discussion is nice and should be added to the manuscript.

---

> > > ### Author Response · Authors · 2022-08-05
> > > **Response to your second run of question**
> > >
> > > Thanks very much for your reply. Static image datasets are also commonly used for efficiency analysis [1,2]. Still, we will do the experiments on Cifar10-DVS and provide the efficiency analysis later. Futher more, the proposed regularizer increased accuracy by increasing the spiking rate indeed. However, the spiking rate is derived by our theory. Obviously, increasing the spiking rate to 100% will decrease accuracy.
> > >
> > > [1] Rathi N ,  Roy K . DIET-SNN: A Low-Latency Spiking Neural Network with Direct Input Encoding & Leakage and Threshold Optimization[J]. Institute of Electrical and Electronics Engineers (IEEE), 2021.
> > >
> > > [2] Deng S ,  Li Y ,  Zhang S , et al. Temporal Efficient Training of Spiking Neural Network via Gradient Re-weighting[J].  ICLR 2022.

---

> ### Author Response · Authors · 2022-08-02
> **Response to Reviewer jVTP (Part IV)**
>
> Here, we still wish to provide some discrepancies between our method and finite difference method [2][5]. First, the finite difference method is very time-consuming, which has been reported in [5] and can be deduced from the description in [2] as we analyzed in A2. Second, the derivation of finite difference can be traced back to the directional derivative and difference in mathematics. Such finite difference is reasonable to be used to approximate the instantaneous derivative at a certain point in ANNs, which have continuous and differentiable activations. The high-similarity curves in experiment 1 in [2] verify this conjecture. However, the finite difference technique is not suitable for estimating the derivative of holder-continuous and non-differentiable functions like SNNs, which have uncontinuous and non-differentiable activations. There may be a large gap between the instantaneous and average derivatives. Thus, the $\epsilon$ should be selected elaborately, which also be verified in figure 2 in [2]. Third, could we use a small part of gradients of parameters to represent all gradients of parameters like [2] to find the optimal gradient may also need further discussions. Nevertheless, we are not denying the soundness and novelty of the finite difference method [2][5], but meaning that scientific research needs different views and arguments.
>
> We can understand that you may wish to find the optimal changing strategy and our present strategy is not optimal in your opinion. That is true that we cannot ensure our strategy is optimal. However, neither [2] nor [5] can ensure that their strategies are optimal with much more computation as analyzed above. Both the two methods provide a possible scheme for the optimal strategy, and both the two methods are a relatively suitable methods from experiments. While we think that, providing a simple and relatively good scheme for designing SGs is also an important contribution in the SNN field.

---

> ### Author Response · Authors · 2022-08-02
> **Response to Reviewer jVTP (Part III)**
>
> Q4: The heuristic description of the ESG is also questionable. How do the authors evaluate the "weight updating ability"? I understand that too sharp a surrogated gradient function cannot sustain training, but the function is also not "the wider the stronger". Too wide the function is also harmful for training. Besides, it is also dubious to claim adjusting the function more and more accurate benefits the training of SNNs. Please refer to discussions in [2] [4] [5].
>
> A4: Sorry for this confusion. We presented in the original paper that most prior works adopted fixed surrogate gradients (SG) to overcome the non-differentiability challenge (see line 148). It is difficult to find a suitable fixed SG, which is also verified by [4] in Section 2.3 that ``Surrogate Gradient Learning Is Sensitive to the Scale of the Surrogate Derivative.'' We noticed that the SG methods are also adopted in Quantization Neural Networks (QNNs) which also suffer from non-differentiability. Some changeable surrogate gradient methods that can achieve better performance by changing SG in the training are proposed in QNNs (see line 150-153). Then We suppose that designing a suitable ESG can also increase accuracy in SNNs(see line 220). This is our motivation which is much different from [2], [4], and [5].
>
>  From experiments, we find that a wide surrogated gradient function can enjoy a fast convergence (see figure 4), i.e., a strong weight updating ability. From figure 3 we can also see that a wide surrogated gradient function keep some relatively large derivative in a relatively large range, which means most parameters can be updated sufficiently in backward phase, i.e., a strong weight updating ability. On the contrary, a sharp surrogated gradient function will lose the ability to update most parameter, since most parameters will suffer a very small gradient in this situation, thus they cannot be updated sufficiently. An extreme example is using the real gradients of firing function, where the network will completely lose the ability to update parameters. Hence we said that a wide surrogated gradient function enjoys a strong weight updating ability. On the other hand, it does not mean ``the wider the better'', since the wider the more inaccurate gradient. it is like that in DNNs, a big learning rate is helpful for fast convergence, but a too big learning rate will be harmful for training.
>
>  Also, we don't mean that a sharp surrogated gradient function can ensure a high accuracy. As we analysed above, a sharp surrogated gradient function will lose the strong weight updating ability, which is also important to the accuracy. Experiments in figure 4 can also verify this phenomenon. Nevertheless, the problem we focus is not whether a sharp surrogated gradient function or a wide surrogated gradient function better for training SNNs, but how to leverage the two important factors in training DNNs, i.e., strong weight updating ability and accurate gradien, since it is not easy to find a fixed suitable surrogated gradient function that can well balance these two factors. Just like that it is not easy to find a fixed suitable learning rate for DNNs. Too big or too small are both not good. But one adopting the dynamic strategy, where a relatively big learning rate at the beginning and a relatively small learning rate in the end will be a more suitable and better solution.
>
> Obviously, the strong weight updating ability is more important at begin due to that all parameters can be updated sufficiently, while the accurate gradient is more important at end due to that the network can obtain the accurate optimization direction. In the same way, we adopt the dynamic strategy that a relatively wide surrogated gradient function at begin and a relatively sharp surrogated gradient function to make the SNN training more suitable, since in this situation the strong weight updating ability at begin and accurate gradient at end are both kept.
>
> Our contribution is that providing a dynamic strategy for surrogated gradient function and it performing better and can be easier to be obtained than a fixed surrogated gradient function in our experiments. Furthermore, the method is very simple and easy to be embedded in other works. From our analysis and experiments, we think our claim of using a dynamic strategy is better than a fixed strategy is convincing. Our observation from experiments that the parameters are learned sufficiently on the premise, adjusting the function to a more accurate gradient benefits the training of SNNs is also convincing.

---

> ### Author Response · Authors · 2022-08-02
> **Response to Reviewer jVTP (Part II)**
>
> About soundness.
>
> Q3: The theoretical reasonsthe authors proposed that lead to the IM-loss are questionable. First, information loss is necessary for classification tasks. Therefore we do not know whether maximizing informaton flow benefits the performance. Second, the measurement of information flow neglects the important temporal dependency of LIF neurons, so the result that p(0)=p(1)=0.5 maximizes the information flow based on elementary information theory is oversimplified. Third, the membrane potential actually does not follow the Gaussian dist with free to change mean value. The paper cites [3] here to back up their claim, yet [3] only mentioned that the membrane potential distribution approximately follows a zero mean Gaussian distribution.
>
> Thanks for your time in reviewing our paper. From your assessments and questions, we can see your professionalism and much effort in checking our work and other related work carefully. Your assessments and questions also give much insights in SNN field. Here, we want to further elaborate more on our work.
>
> First, information loss is necessary for classification tasks indeed. However, the information loss in activation layer is disgusted. Information loss means the information cannot be fully reconstructed. There are two important modules in networks: convolution/fully-connected layer and activation layer. The convolution layer is an irreversible transformation, thus cannot be reconstructed. It usually compresses out non-relevant information and leaves useful information by learning. However, the activation layer usually plays the role of nonlinear transformation to make the network more complex. More deeper, more complex. At the same time, the information loss in activation layer is disgusted. A powerful proof is that PReLU[7] performs better than ReLU, where PReLU is a reversible transformation, thus without information loss, while ReLU is a irreversible transformation, thus with information loss. The form of PReLU is learned in the training. It may also be learned to be ReLU, while not in experiments. This can prove that activation layer without information loss is better. From another view, the activation function is usually fixed in DNNs without learning in the network. A fixed form is not easy designed to play such a role that compressing out non-relevant information and leaving useful information. Indeed, ReLU will introduce information loss but still be widely used. However, the information loss for ReLU is acceptable and ReLU dropping the negative part also plays a role of ``dropout". Still ReLU dropping the negative part mainly want to retain the nonlinear transformation of network not to abandon information. However, the firing function of the SNN is an irreversible transformation with serious information loss, since it forces all information only to two values. In the work, we focus on solving information loss in activation layer. It doesn't conflict with your views that information loss is necessary for classification tasks, since this information loss you referred mainly comes from convolution layer not activation layer.
>
> Second, we also advocate for your view that the information flow is temporal dependent. However, this will still not change the fact the information flow will be maximized when  p(0)=p(1)=0.5 based on the information theory we adopted. Though the p(0)=p(1)=0.5 is seen as simple, it is the result derived from the theory but not the implementation manner of the theory. Considering the temporal dependency of LIF neuron is very complex and difficult to express explicitly and uniformly, we choose optimizing spiking rate only based on the information theory and do not design any explicit temporal dependency module in the IM-Loss. In this way, their dependency can be learned freely and implicitly in the training instead of manual design beforehand. This is like we do image classification task with learned CNNs instead of handcrafted descriptors. Furthermore, this design is very simple and general thus easy to be applied to other models. On the other hand, we do not mean embed the temporal dependency explicitly in the regularization is not good, but learning it implicitly and simply by learning manner is also a feasible scheme.
>
> Third, I'm sorry for this neglect of experiment proof. In our experiment, the membrane potential indeed follow the Gaussian-alike distribution with free mean value. We provide this experiment proof in the appendixes section of revised manuscript for you checking. tdBN[3] also follow the Gaussian-alike distribution with free mean value in our experiments and other work (see Fig. 2 in [8]).
>
> [7] He, K. , et al. "Delving Deep into Rectifiers: Surpassing Human-Level Performance on ImageNet Classification." CVPR IEEE Computer Society, 2015.
>
> [8] Guo, Y., Tong, X., , et al. Recdis-snn: Rectifying membrane potential distribution for directly training spiking neural networks. In:  (CVPR2022).

---

> > ### Comment · Reviewer_jVTP · 2022-08-05
> > **Respond to Part II**
> >
> > Thanks to the authors for taking the effort to address my concerns. However, I still hold my view.
> >
> > 1. The authors admit that the information loss is necessary, but then suggest that information loss in activation layers is, on the contrary, disgusted. This is quite confusing. It is also controversial whether PReLU or leaky ReLU always outperforms ReLU (the most prevalent choice in ANN training).
> >
> > 2. The authors suggest that temporal dependency will not change the fact the information flow will be maximized when p(0)=p(1)=0.5. But they do not answer the reason why. The response only mentions that considering temporal dependency is very complex and difficult.
> >
> > 3. Thanks for your additional figure. However, it further confirmed my view: the mean value of membrane potential is *not* free to change. Despite the regularization term the author proposed to drive the mean value towards threshold 1, the resulting distribution has a mean value of 0.5. The resetting mechanism of the LIF neurons makes it hard to have half membrane potential lies above the threshold as expected. Therefore, the information flow is also not maximized as the author expected.

---

> > > ### Author Response · Authors · 2022-08-05
> > > **Response to your second run of question**
> > >
> > > A1: Thanks very much for your reply. We has confirmed that the information loss for ReLU is acceptable and ReLU dropping the negative part also plays a role of ``dropout". Still ReLU dropping the negative part mainly want to retain the nonlinear transformation of network not to abandon information. Unfortunately, it can't convince you.
> > >
> > > Here, we want to provide you more prior works in BNNs, which also binarizes the activations even weights, to verify our statement. Liu et al. [1] thought binary activations will induce information loss and added extra shortcut between consecutive convolutional blocks to strengthen the representational capacity of the network (see on the third paragraph of page 2). [2][3] proposed to compensate to some extent for the quantization error (i.e. information loss, we also point that quantization of the spike activity function will cause information loss (see line 162)) caused by the binary approximation by re-scaling the output of the binary convolution using real-valued scale factors. (see on the second paragraph of page 2 of [2]).
> > >
> > >
> > > [1] Zechun Liu, Baoyuan Wu, Wenhan Luo, Xin Yang, Wei Liu, and Kwang-Ting Cheng. Bi-real net: Enhancing the performance of 1-bit cnns with improved representational capability and advanced training algorithm. In ECCV, pages 722–737, 2018.
> > >
> > > [2] Bulat A ,  Tzimiropoulos G . XNOR-Net++: Improved Binary Neural Networks[J].  2019.
> > >
> > > [3] Rastegari, Mohammad , et al. "XNOR-Net: ImageNet Classification Using Binary Convolutional Neural Networks." Springer, Cham (2016).
> > >
> > >
> > > From another view, the activation function is usually fixed in DNNs without learning in the network. A fixed form is not easy designed to play such a role that compressing out non-relevant information and leaving useful information.
> > >
> > > The firing function of the SNN is an irreversible transformation with serious information loss, since it forces all information only to two values. If we think the information loss is acceptable, an extreme example is one can force all information to 1 value. Obviously, the network will lose its classification ability completely in this situation.
> > >
> > > ---
> > >
> > > A2: Ideally, the spike tensor O should reflect the information of the membrane potential tensor U as much as possible. In the view of the information flow, to maximize the information flow from the full-precision tensor U to the binary tensor O, the mutual information I(U ; O) of the random variables U and O should be maximized. Then p(0)=p(1)=0.5 is derived by information entropy theory.
> > >
> > > ---
> > >
> > > A3: Sorry for this confusion, the firing threshold is 0.5 in the paper (see figure 3 or our provided code in line 11 of layer file). The resulting distribution has a mean value around 0.5 too.

---

> > > > ### Comment · Reviewer_jVTP · 2022-08-05
> > > > **3nd Round Response**
> > > >
> > > > Thanks for your timely response.
> > > > A1 & A2. Still, I am not fully convinced. Regarding minimizing quantization error, one should let U in [-inf, threshold/2] be as close to 0 as possible, and let U in [threshold/2, +inf] be as close to the threshold as possible. Why does it equivalent to maximizing information flow? How do you evaluate the difference between these two approaches? I am willing to increase my overall rating if this question is answered.
> > > >
> > > > A3. Sorry for my misunderstanding. It addresses my concerns. Surprisingly knowing the distribution is not affected by the resetting mechanism.

---

> > > > > ### Author Response · Authors · 2022-08-05
> > > > > **Response to your third run of question**
> > > > >
> > > > > Thanks for your timely response too. We are very grateful to meet such responsible reviewer. Indeed, there are some work that introduces a regularization function to encourage full precision values around binary values[1]. However, there is still work like [2] that added extra shortcut between consecutive convolutional blocks to transmit the full precision values to binary values directly. These all interesting works to reduce information loss. i.e., the methods to reduce information loss can be different.
> > > > >
> > > > > Our method and [1] are not equivalent. [1] reduces the information loss from single thought, i.e., if every neuron will reduce the quantization error, the information loss for the whole model will be reduced, thus the regularization of [1] will add to every precision value. While we reduce the information from monolithic thought. i.e., we analyze the information expressiveness ability of the SNN based on information entropy first and derived the optimal distribution of spikes. Then we add a regularization to the whole spike distribution.
> > > > > Nevertheless, the accuracy increasing mainly corresponds cross-entropy-Loss reducing. i.e., to reduce the information loss is only the manner, to reduce the cross-entropy-Loss is the goal. Ours and [1] both add a new constraint in the network optimization.
> > > > >
> > > > > Since the regularization changes the optimization objective, you may concern that it will lead to a configuration far from the global optimal of the cross-entropy loss function. However, prior theoretical [3,4] and empirical [5] work has shown that a deep neural network can have many high-quality local optima. Kawaguchi proved that under certain conditions, every local minimum is a global minimum [4]. Through experiments, Im et al. show that using different optimizers, the achieved local optima are very different [5]. These insights show that adding the regularization may deviate the training away from the original optimal, but can still lead to a new optimal with high accuracy. Moreover, the regularization diagnoses the poor conditions of the activation flow, and therefore may achieve higher accuracy. Our experiment results confirm this hypothesis.
> > > > >
> > > > >
> > > > > [1] Darabi S ,  Belbahri M ,  Courbariaux M , et al. BNN+: Improved Binary Network Training[J].  2018.
> > > > >
> > > > > [2] Zechun Liu, Baoyuan Wu, Wenhan Luo, Xin Yang, Wei Liu, and Kwang-Ting Cheng. Bi-real net: Enhancing the performance of 1-bit cnns with improved representational capability and advanced training algorithm. In ECCV, pages 722–737, 2018.
> > > > >
> > > > > [3] A. Choromanska, M. Henaff, M. Mathieu, G. B. Arous, and Y. LeCun. The loss surfaces of multilayer networks. In Artificial Intelligence and Statistics, pages 192–204, 2015.
> > > > >
> > > > > [4] K. Kawaguchi. Deep learning without poor local minima. In Advances in Neural Information Processing Systems, pages 586–594, 2016.
> > > > >
> > > > > [5] D. J. Im, M. Tao, and K. Branson. An empirical analysis of deep network loss surfaces. 2016.
> > > > >
> > > > >
> > > > > If you have further questions, please do not hesitate to reply to us. Thanks again.

---

> ### Author Response · Authors · 2022-08-02
> **Response to Reviewer jVTP (Part I)**
>
> Thanks for your time in reviewing our paper. From your assessments and questions, we can see your professionalism and much effort in checking our work and other related work carefully. Your assessments and questions also give us much insights in SNN field. Here, we want to provide you some different perspectives to discuss further.
>
> About Novelty.
>
> Q1: Adding homeostatic mechanisms to regularize spiking neuron's firing activity is not a new trick. Previously, [1] adds a regularizer to drive SNNs' firing activity more sparse to save energy. However in this work, the author proposed regularizer drives membrane potential near threshold, which potentially increases the firing activity and leads to poorer energy efficiency.
>
> A1: Indeed, adding homeostatic mechanisms to regularize spiking neuron's firing activity is not a new trick. We admire [1]'s work in adding homeostatic mechanisms to regularize spiking neuron's firing activity. However, our motivation comes from the issue of information loss that we notice in SNNs. Then we provide a new perspective/idea of handling such issue by maximizing the information flow in SNNs. In order to achieve this goal, we derive the optimal case of SNN information expression based on information theory. Finally we resort to regularize loss to find the optima. Our motivation, idea, perspective, and logical thought are different. Even our regularization method is much different from [1]. In this perspective, we think our novelty is sufficient. We will provide more soundness about information loss of SNNs and efficiency comparison in later questions.
>
> ---
>
> Q2: Updating the shape of the surrogated function is also not new. [2] proposed to automatically adjust the temperature of the surrogated activation function in order to achieve the best training performance. Comparing to this work, [2] is more flexible.
>
> A2: Indeed, updating the shape of the surrogated function is also not new. However, to realize this idea, both [2] and ours adopt different methods. Here, we want to provide some discrepancies between the two methods. Our method is designed manually, and in our experiments, the SNN with ESG is better than the vanilla SNN but will induce no extra computation in the training. While the Dspike[2] is computed by finite difference method, however, evaluating finite difference could be time-consuming, since for each single weight, the model should run two times to evaluate the difference of the loss, and a model can have more than ten million parameters (e.g. 11,200,000 parameters for ResNet18), which greatly slows down the training process. To reduce the computation, the Dspike [2] chooses to only compute the finite difference in the first layer to represent the surrogate gradients of the whole model. However, this is still very time-consuming. Take ResNet20 in Cifar10 for example as introduced in [2], if we set batch-size as 128. The computation for the finite difference in the first layer once is equal to ResNet20 inferring about 4.5 epochs on training set. To sum up, the Dspike performs remarkably while its training is very time-consuming. Our method is more efficient with a relatively good performance. Hence, we think ESG function and Dspike function are both meaningful for SNN field. This is some like that SGD optimizer is designed manually from experience, while Meta-optimizer is learned [6] in the training. Though Meta-optimizer [6] performs better than SGD optimizer, it is time-consuming and more complex to use. And it cannot be concluded that Meta-optimizer is better and more flexible than SGD optimizer.
>
> [6] Andrychowicz M ,  Denil M ,  Gomez S , et al. Learning to learn by gradient descent by gradient descent[C], Proceedings of the 30th International Conference on Neural Information Processing Systems, 2016.

---

### Official Review · Reviewer_4Mt9 · 2022-07-11

**Rating:** 6
**Confidence:** 4
**Soundness:** 3 good
**Presentation:** 3 good
**Contribution:** 3 good

**Summary:**

The paper proposed the information maximization (IM) loss for training deep SNN which maximizes information flow of the network and indirectly provides normalization during training. The IM loss is constructed to strengthen the mutual information between the membrane potential and the following spiking activity. In addition, the ESG method is propossed to improve network training. Experiments on benchmark image classification tasks show that the method improves the accuracy of SNN, and on CIFAR10 it surpass current SOTA SNN.

**Questions:**

The ESG method seems quite intuitive and empirical, is there any theoretical justification that can help to support it?


**Limitations:**

It would be helpful if the potential negative impact of moving membrane potential towards the spiking threshold can be discussed.
All SOTA SNN works should be listed on table to avoid misleading.

**Strengths And Weaknesses:**

The work proposed the IM loss to reduce information loss from membrane potential to spike, which moves the mean membrane potential towards the spiking threshold during training. The idea is new for the training of SNN and the soundness of the approach is sufficient, based on the derived equal relation between the mutual information (MI) of the membrane potential and the spike and the entropy of spiking distribution. However, driving the mean membrane potential near the threshold becomes an inductive bias, which could be suboptimal for the network performance under certain situation. Another outcome of the approach is that it increases the activation of the resulting network, which impairs the sparse computation advantage of SNN.
The ESG function is similar to the Dspike function (NeurIPS 2021, Yuhang Li, et. al), which should be cited.
In the experiment, except CIFAR10, the approach is outperformed by current SOTA SNN (e.g. Dspike (2021 NeurIPS), TET (2022 ICLR) have higher accuracies on CIFAR100, ImageNet and CIFAR10-DVS, though not listed on the table), leaving the general usage of the method questionable.

---

> ### Author Response · Authors · 2022-08-02
> **Response to Reviewer 4Mt9 (part II)**
>
> Q3: The ESG method seems quite intuitive and empirical, is there any theoretical justification that can help to support it?
>
> A3: Indeed, the ESG method design is intuitive and empirical like these learning rate design in DNNs. It also comes from our experiments and understandings in SNN field. However, we designed the ESG method form meticulously and carefully. From large amount of experiments and thoughts, we find two RULEs of designing K(i). First, it should have a growing trend. As explained in Section 4.3, using EvAF with a smaller k results in the SNN with strong weight-updating ability; while a larger k results in accuracy gradients. To obtain both weight updating ability in the early stage of training and accurate backward gradient at the end of the training, K(i) should have a growing trend. Second, it should enjoy long-term maintenance of weight updating ability. As shown in Fig. 4 in the paper, due to the stronger weight-updating ability, SNNs with the EvAF using a fixed smaller k are much easier to converge to better results than these using a fixed larger k. This means EvAF with smaller k values taking up more training time is better. According to these rules, we choose K(i) with exponential growth rather than other linear or logarithmic growth.
>
> ---
>
> Q4: It would be helpful if the potential negative impact of moving membrane potential towards the spiking threshold can be discussed.
>
> A4: Thanks for your constructive advice. In our all experiments, driving the mean membrane potential near the threshold is better than doing nothing. However, as you said, driving the mean membrane potential near the threshold becomes an inductive bias, which could be suboptimal for the network performance under certain situation. Though the suboptimum is still better than the vanilla results, a better choice may be driving the mean membrane potential to a range. This suppose needs sufficient theory and experiments support. Nevertheless, our this work still provides a new perspective to understand and solve the information loss of SNNs and two simple and solid methods in this field.

---

> ### Author Response · Authors · 2022-08-02
> **Response to Reviewer 4Mt9 (part I)**
>
> Thanks for your efforts in reviewing our paper and your recognition of our idea to maximize mutual information and soundness of the IM-Loss approach. Some other confusions we also want to explain piece by piece as followed.
>
> ---
>
> Q1: The ESG function is similar to the Dspike function (NeurIPS 2021, Yuhang Li, et. al), which should be cited.
>
> A1: Thank for your kind reminder. We will cite (NeurIPS 2021, Yuhang Li, et. al) in the revised version. Here, we want to provide some discrepancies between these two methods. The ESG function is designed manually, and in our experiments, the SNN with ESG is better than the vanilla SNN but will induce no extra computation in the training. While the Dspike is computed by finite difference method, however, evaluating finite difference could be time-consuming, since for each single weight, the model should run two times to evaluate the difference of the loss, and a model can have more than ten million parameters (e.g. 11,200,000 parameters for ResNet18), which greatly slows down the training process. To reduce the computation, the Dspike chooses to only compute the finite difference in the first layer to infer the surrogate gradients of the whole model. However, this is still very time-consuming. To sum up, the Dspike performs remarkably while is very time-consuming. Taking ResNet20 in Cifar10  (NeurIPS 2021, Yuhang Li, et. al) as an example, if we set batch-size as 128. The computation for the finite difference in the first layer once is equal to ResNet20 inferring about 4.5 epochs on training set. Our method is more efficient with a relatively good performance. Hence, we think ESG function and Dspike function are both meaningful for SNN field.
>
> ---
>
> Q2: In the experiment, except CIFAR10, the approach is outperformed by current SOTA SNN (e.g. Dspike (2021 NeurIPS), TET (2022 ICLR) have higher accuracies on CIFAR100, ImageNet and CIFAR10-DVS, though not listed on the table), leaving the general usage of the method questionable.
>
> A2: Thank for your very kind reminder. We will cite and compare these papers you mentioned in the revised version. For Dspike, it indeed performs better than our method in most datasets, however, it is much more time-consuming than our method. For TET, actually our method performs better on most datasets, such confusion comes from that we used different model architectures in the original papers. Here, we provide the comparison with same architectures bellow. For CIFAR-100, we used the same backbone ResNet19 as TET. It can be seen that our model achieved 78.51\% top-1 accuracy with 6 timesteps, which outperforms its TET counterpart with 3.79\% higher accuracy. For the ImageNet dataset, results we reported in original paepr show that our Spiking ResNet34 achieves a 2.64\% increment on TET spiking ResNet34. The accuracy of our ResNet34 does not exceed TET SEW ResNet34. However, TET SEW ResNet34 transmits information with integers, which is not a typical SNN.
>
> |Dataset|Method|Architecture|Timestep|Spike form|Accuracy|
> |---|---|---|---|---|---|
> |CIFAR100|TET|ResNet-19|6|Binary|74.72%|
> |CIFAR100|TET|ResNet-19|4|Binary|74.47%|
> |CIFAR100|TET|ResNet-19|2|Binary|72.87%|
> |CIFAR100|Our method|ResNet-19|6|Binary|78.51%|
> |CIFAR100|Our method|ResNet-19|4|Binary|77.42%|
> |CIFAR100|Our method|ResNet-19|2|Binary|74.56%|
> |ImageNet|TET|Spiking-ResNet34|6|Binary|64.79%|
> |ImageNet|TET|SEW-ResNet34 |4|Integer|68.00%|
> |ImageNet|Our method|Spiking-ResNet34|6|Binary|67.43%|

---

> > ### Comment · Reviewer_4Mt9 · 2022-08-09
> > **Thanks for the response**
> >
> > I thank the authors for their reponse and clarifications to my questions. I still holds the concerns on maximizing the information flow in SNN will possibly reduce the sparseness the SNNs and hence largely decrease their practicability.

---

> > > ### Author Response · Authors · 2022-08-09
> > > **Response to your second run of concerns**
> > >
> > > Thanks very much for your reply and kind reminder. we are very sorry for this neglect. Reviewer jVTP presented the same concern and we has provided the response in A5 for him/her . But we are very sorry for this neglect for you again. Here, we provide the efficiency analysis with tdBN[1] which is commonly used in SNNs. We define the average spike rate as total #spikes/#neurons on the test-set. For tdBN, the avg-rate of ResNet-19 on CIFAR10 are 0.35, 0.67, 0.82 with T = 2, 4, 6. while 0.81, 1.59, 2.37 for IM-Loss. However, with only T=2, our method outperforms the tdBN with T=6 by 0.69% accuracy with a low avg-rate. IM-Loss increases the fire rate but reduces the timesteps at the same time, i.e., the proposed regularizer can achieve a better trade-off between spiking sparsity and accuracy. On the other hand, SNNs can also run on GPUs. In this situation, the fewer timesteps, the better, and our method enjoys very few timesteps.
> > >
> > > [1] H. Zheng, Y. Wu, L. Deng, Y. Hu, and G. Li. Going deeper with directly-trained larger spiking neural networks. arXiv,10 2020.

---

> > > > ### Comment · Reviewer_4Mt9 · 2022-08-09
> > > > **response to authors on the sparseness of spikes**
> > > >
> > > > Thank you for clarification and this additional experiment. I personally encourage the authors to include more discussions in this respect on the revised manuscript (including positive and negative aspects). Since running on GPUs for SNN is not the ultimate goal (which I believe will not bring significant advantages in terms of efficiency), running on specific hardwares(e.g. neuromorphic chips) that fits SNN's event-driven mechanism and sparseness should eventually reveal it's high efficiency. In this case, the overall firing rate becomes crucial.

---

> > > > > ### Author Response · Authors · 2022-08-09
> > > > > **Response to your third run of concerns**
> > > > >
> > > > > Thanks for your timely reply. We are very grateful to meet such a responsible reviewer. We have taken your suggestion seriously and added an extra section to discuss the sparsity and efficiency cost in the revised version (see line 300-308 in revision). We'd like to know if you have any other questions. We would be very grateful that if you can re-consider your rating on the premise that we could address your concerns with our best efforts.

---

### Official Review · Reviewer_t7Yt · 2022-07-12

**Rating:** 5
**Confidence:** 4
**Soundness:** 3 good
**Presentation:** 3 good
**Contribution:** 2 fair

**Summary:**

The paper proposes an IM loss based training method for SNNs that shows high accuracy at low latency.

**Questions:**

Please see my above comments on weakness and clarify teh technical novelty of the work.

**Limitations:**

Please see weakness section comments.

**Strengths And Weaknesses:**

+The paper's experiments shows that the IM loss trains SNNs with SOTA results compared to works cited by the author.
-The paper presents a direct training method using BP for SNNs. This work is very derivative and incremental. There is a lot of work from Priya Panda's group at Yale, Emre Neftci's group, and many others with regard to SNN training. The authors have failed to acknowledge most recent works and the method they are proposing is very incremental in the context of those works. Further, many recent works on SNNs have targeted larger datatsets including video segmenattion with direct training.
-Recent works like [5] show SNN training at very low latency with novel architectures. [10] uses a BN rule to train temproal SNNs. Is there a relationship between the implicit IM loss normalizationa nd the expicit normalization employed in [10]?

Below is a list of publications (not exhaustive) that the author should check:

[1] Towards spike-based machine intelligence with neuromorphic computing K Roy, A Jaiswal, P Panda Nature 575 (7784), 607-617

[2] Enabling spike-based backpropagation for training deep neural network architectures C Lee, SS Sarwar, P Panda, G Srinivasan, K Roy Frontiers in neuroscience, 119

[3] Rate Coding Or Direct Coding: Which One Is Better For Accurate, Robust, And Energy-Efficient Spiking Neural Networks? Y Kim, H Park, A Moitra, A Bhattacharjee, Y Venkatesha, P Panda ICASSP 2022-2022

[4] Neuromorphic Data Augmentation for Training Spiking Neural Networks Y Li, Y Kim, H Park, T Geller, P Panda arXiv preprint arXiv:2203.06145

[5] Neural architecture search for spiking neural networks Y Kim, Y Li, H Park, Y Venkatesha, P Panda arXiv preprint arXiv:2201.10355

[6] Optimizing deeper spiking neural networks for dynamic vision sensing Y Kim, P Panda Neural Networks 144, 686-698

[7] Federated Learning with Spiking Neural Networks Y Venkatesha, Y Kim, L Tassiulas, P Pand IEEE Transactions on Signal Processing 2021

[8] Beyond classification: directly training spiking neural networks for semantic segmentation Y Kim, J Chough, P Panda arXiv preprint arXiv:2110.07742

[9] Visual explanations from spiking neural networks using interspike intervals Y Kim, P Panda Scientific Reports 11, Article number: 19037 (2021)

[10] Revisiting batch normalization for training low-latency deep spiking neural networks from scratch Y Kim, P Panda Frontiers in neuroscience, 1638

---

> ### Author Response · Authors · 2022-08-02
> **Response to Reviewer t7Yt**
>
> Thanks for your time in reviewing our paper and we are gratitude to have your recognition of our work and experiments. Your constructive advice for introducing and analyzing the prior excellent works comprehensively will make our paper more objective and substantial. Actually, we have also noticed some other very excellent work with extraodinary contributions in SNN field like [11] before and prepared to acknowledge in the camera-ready. Thanks for your constructive suggestion again and we think these important works should be comprehensively introduced. This suggestion will not only benefit for this revised version but also our future papers.
>
> For the relationship and difference between BNTT[10] and IM-Loss, with adjusting the pre-activation, they both can make the SNN deeper and more accurate. However, BNTT start from finding a more suitable BN for SNN, while IM-Loss starts from reducing information loss of SNN based on information theory. Hence their implementation manners are very different. BNTT performs better than original BN in SNNs. While IM-Loss plays a part of the role of BN but without introducing any multiply-and-accumulate (MAC) operations in the inference phase as BN.
>
> Considering your recognition of our work and the high evaluation that ``This work is very derivative and incremental.'', and the neglect of introducing and analyzing the prior excellent works comprehensively can be improved in the revised version as your rich suggestions, we really appreciate it if you can kindly re-check our valuable insights, ideas, methods, and contributions and re-consider your ratings.
>
> [11] Lottery Ticket Hypothesis for Spiking Neural Networks, Youngeun Kim, Yuhang Li, Hyoungseob Park, Yeshwanth Venkatesha, Ruokai Yin, Priyadarshini Panda, ECCV2022.

---

### Official Review · Reviewer_DHw2 · 2022-07-21

**Rating:** 6
**Confidence:** 4
**Soundness:** 3 good
**Presentation:** 2 fair
**Contribution:** 3 good

**Summary:**

The manuscript tackles the problem of supervised learning in spiking neural nets (SNNs). The authors claim that SNNs loose information in feed-forward pass because of the binary nature of their responses and propose an information-maximizing loss function to reduce such loss.
For gradient back-propagation they propose to extend the popular surrogate gradient approach with a surrogate gradient who's spike activity function changes during the training. In multiple experiments the authors demonstrate advantages of their approach.

**Questions:**

Section 4.2 is confusing. If the spikes can not reach the output as shown in figure 2, how the SNNs can be trained at all?

The dependence of results on the integration time of outputs is stated only for selected numbers (lines 283-291). It would be helpful to instead plot them as in figure  4. Without these plots these results seem as cherry-picked.



**Limitations:**

The authors did not address limitations of their approach.

**Strengths And Weaknesses:**

Strengths. The authors propose two interesting extensions of existing surrogate backprop learning method for SNNs. The first one, infomax-inspired loss, maximizes the entropy of individual neuronal outputs under the assumption of independent neuronal responses. This is a valuable insight that SNN community may find of interest. The second one, using a schedule to adjust the activation function is quite innovative. The experiments with two SNN architectures and ablations provide compelling evidence that the proposed approach has advantages over previous work.
Weaknesses. While the aforementioned ideas are somewhat innovative, the implementation is quite limited. For example, the infomax assumes that the network outputs are independent, thus optimizing only spiking rate. Moreover, the "Evolutionary Surrogate Gradients" is essentially a manually set schedule for adjusting a (hyper)parameter, gradient width, with training time. THe form of ESG change with time is hand-picked and no experiments are offered to shed light as to why this form is preferred and how it compares with alternative schedules.
I also found the term "evolutionary" highly confusing as the approach has nothing to do with evolutionary optimization, and only referes to the aforementioned manually selected dependence of gradient width on time. A much better term would be "adjustable, train time-dependent" or alike.

---

> ### Author Response · Authors · 2022-08-02
> **Response to Reviewer DHw2**
>
> Thanks for your precious time for reviewing our work and for recognition of our ideas for the loss design and surrogate gradient design. Here we will first respond to your comments piece by piece and then provide more deep thoughts about methods with respect to your concerns of our implementation.
>
> ---
>
> Q1: Section 4.2 is confusing. If the spikes can not reach the output as shown in figure 2, how the SNNs can be trained at all?
>
> A1: Sorry for this confusion. The SNN without any normalization techniques or our IM-Loss is difficult to train since the spikes can not reach the output. With IM-Loss, the firing rate can be adjusted layer by layer until the spikes reach the output. More specifically, if the spikes could only reach some middle layer at the beginning, the firing rate of the middle layer would be rather low and turn to be 0 in its next layer. Since the IM-Loss accumulates the information loss of all the layers (Eq.11 in the original paper), it can directly optimize each layer's firing rate in the backward phase instead of gradually going back from the last layer to the first layer like the cross-entropy loss. When the IM-Loss adjust the middle layer's firing rate to a certain extent, its next layer will receive sufficient spikes and then fire spikes to the later layer. Such behavior will continue until the spikes reach the output layer. Similarly, the normalization technique can also change the firing rate of each layer directly instead of gradually going back from the last layer to the first layer.
>
> ---
>
> Q2: The dependence of results on the integration time of outputs is stated only for selected numbers (lines 283-291). It would be helpful to instead plot them as in figure 4.
>
> A2: Thanks for your constructive advice. More detailed results are as follows and we will plot them in the revised paper.
>
> | Method  | Architecture | Timestep | Accuracy |
> |----|----|----|----|
> |Diet-SNN|VGG-16|5|92.70%|
> |STBP-tdBN|ResNet-19|6|93.16%|
> |STBP-tdBN|ResNet-19|4|92.92%|
> |STBP-tdBN|ResNet-19|2|92.34%|
> |Our method|VGG-16|6|94.01%|
> |Our method|VGG-16|5|93.85%|
> |Our method|VGG-16|4|93.52%|
> |Our method|ResNet-19|6|95.49%|
> |Our method|ResNet-19|5|95.50%|
> |Our method|ResNet-19|4|95.40%|
> |Our method|ResNet-19|3|94.96%|
> |Our method|ResNet-19|2|93.85%|
> |Our method|ResNet-19|1|92.01%|
>
> ---
>
> For the concern of “implementation novelty”, we want to explain and clarify more in the following few aspects:
>
> In terms of the method design principle. We want to design a simple but effective method as our other works. 1) We think with effective but simpler implementation/method, the correctness and validity of our ideas could be verified more convincingly. With correct and meaningful ideas, more following works can be inspired and more useful methods can be provided. This is also one of our original intentions and contributions. 2) The simpler, the more versatile. We wish our methods can be easily reproduced and applied to other methods without adding any burden. Then our methods may have a chance to be a followable benchmark for the future works. Based on this principle, we designed these too simple but effective methods.
>
> In terms of the technical implementation. For IM-Loss, we advocate for your viewpoint that the network outputs are dependent. However, the dependency are difficult to express explicitly and uniformly, hence we choose optimizing spiking rate only based on the information theory. In this way, their dependency can be learned freely in the training instead of manual design beforehand. This is like we do image classification task with learned CNNs instead of handcrafted descriptors. On the other hand, this design is very simple and general, thus easy to be embedded in other work. For ESG, we have also been thinking about how to find a training-aware manner. However, it is not an easy work. The EvAF is only used in the backward but not in the forward. It does not involve in the loss calculation (can not be trained). Then we turn to find a simple but effective manual manner. From methodology and experimental analysis, we find two RULEs of designing K(i). First, it should have a growing trend. As explained in Section 4.3, using EvAF with a smaller k results in the SNN with strong weight-updating ability; while a larger k results in accuracy gradients. To obtain both weight updating ability in the early stage of training and accurate backward gradient at the end of the training, K(i) should have a growing trend. Second, it should enjoy long-term maintenance of weight updating ability. As showed in Fig. 4 in the paper, due to the stronger weight-updating ability, SNNs with the EvAF using a fixed smaller k are much easier to converge to better results than these using a fixed larger k. It means that it is better to take up more training time for the EvAF with smaller k values. According to these rules, we choose K(i) with exponential growth rather than other linear or logarithmic growth.

---

### Author Response · Authors · 2022-08-05
**Hoping to discuss with the reviewers**

Dear Chairs and Reviewers,

We are writing this letter to show our hope of having discussions with you. We received the borderline accept average rating at pre-rebuttal period, thus a further in-depth discussion will help the AC make a fair judgment.

We appreciate all your precious time for reviewing our work. There are two major concerns. 1) For information loss, we focus the information loss in activation layer, which is not like convolution/fully-connected layer that compresses out non-relevant information and leaves useful information by learning. The information loss in activation layer of SNNs should be avoided. 2) For Evolutionary Surrogate Gradient, it is very efficient, while DSpike is very time-consuming. We give detailed proofs and explanations in responses.

Could you please take a little bit of time to read our responses and discuss with us? Thank you all!

Best regards.

---

### Meta-Review · Area_Chair_4CYo · 2022-08-24

**Recommendation:** Accept
**Confidence:** Certain

**Metareview:**

This paper proposes a novel loss for training a spiking neural network that mitigates errors due to quantization. All reviewers agreed that the contributions of this paper were above the acceptance threshold.

**Award:**

No

---

### Decision · Program_Chairs · 2022-09-14

Accept